# Silver–Gold Alloy Nanoparticles (AgAu NPs): Photochemical Synthesis of Novel Biocompatible, Bimetallic Alloy Nanoparticles and Study of Their In Vitro Peroxidase Nanozyme Activity

**DOI:** 10.3390/nano13172471

**Published:** 2023-09-01

**Authors:** Prakash G. Kshirsagar, Valeria De Matteis, Sudipto Pal, Shiv Shankar Sangaru

**Affiliations:** 1Department of Biochemistry and Molecular Biology, University of Nebraska Medical Center, Omaha, NE 68198, USA; 2Department of Mathematics and Physics “Ennio De Giorgi”, University of Salento, 73100 Lecce, Italy; valeria.dematteis@unisalento.it; 3Department of Innovation Engineering, University of Salento, 73100 Lecce, Italy; sudipto.pal@unisalento.it; 4Saudi Aramco, Dhahran 31311, Saudi Arabia

**Keywords:** silver nanoparticles, gold, nanoparticles, gold silver bimetallic alloy, photoreduction, nanozyme, L-tyrosine, peroxidase activity

## Abstract

Facile synthesis of metal nanoparticles with controlled physicochemical properties using environment-friendly reagents can open new avenues in biomedical applications. Nanomaterials with controlled physicochemical properties have opened new prospects for a variety of applications. In the present study, we report a single-step photochemical synthesis of ~5 nm-sized silver (Ag) and gold (Au) nanoparticles (NPs), and Ag–Au alloy nanoparticles using L-tyrosine. The physicochemical and surface properties of both monometallic and bimetallic NPs were investigated by analytical, spectroscopic, and microscopic techniques. Our results also displayed an interaction between L-tyrosine and surface atoms that leads to the formation of AgAu NPs by preventing the growth and aggregation of the NPs. This method efficiently produced monodispersed NPs, with a narrow-sized distribution and good stability in an aqueous solution. The cytotoxicity assessment performed on breast cancer cell lines (MCF-7) revealed that the biofriendly L-tyrosine-capped AgNPs, AuNPs, and bimetallic AgAu NPs were biocompatible. Interestingly, AgAu NPs have also unveiled controlled cytotoxicity, cell viability, and in vitro peroxidase nanozyme activity reliant on metal composition and surface coating.

## 1. Introduction

Bimetallic alloy nanoparticles (NPs) have received considerable attention because of their unique physicochemical and catalytic properties, which make them useful for diverse applications [1,2,3,4,5]. The heterogeneity in the composition and structure of the alloy NPs can integrate the functions of monometallic nanomaterials, provide a synergistic effect, and enhance efficiency and multi-tasking abilities [6,7,8]. The process of alloying two or more metal components with variable molar ratios can provide great dimension to explore the optical, catalytic, electronic, and magnetic properties, which are significantly different from the corresponding monometallic counterparts [6]. Among all the types of bimetallic alloys, the Ag–Au bimetallic alloy NPs have been extensively investigated, owing to their characteristic plasmonic and catalytic properties [2,6,9,10,11,12]. These nanoalloys have been implemented in diverse fields, such as antibacterial, bioimaging, bio-labeling, catalysis, drug delivery, cancer therapy, nanophotonics, optoelectronics, molecular detection, and nanoscale optical biosensors, etc. [8,13,14,15,16,17,18,19,20,21]. The Ag–Au alloy NPs have demonstrated superior catalytic activity towards the oxidation of CO at low temperatures to monometallic Ag or AuNPs [16,22]. The Au alloys have been utilized as a Pt replacement for oxygen reduction in PEM fuel cells [23] and displayed improved electrocatalytic activity in direct borohydride fuel cells [24]. These exciting physicochemical properties and applications of nanoalloys strongly depend on size, structure, composition, and surface chemistry. Therefore, synthesizing them with the accurately controlled size and shape of the desired compositions, along with enhanced stability and dispersion, is a worthwhile objective.

In the past decade, several synthesis strategies have been developed to fabricate the Ag–Au nanoalloys of the desired size in aqueous, organic, and other media. This includes capillary micro-reaction, co-reduction, sonochemical, thermal decomposition, laser ablation in liquid, radiolytic synthesis, citrate reduction of Ag and Au salts, the photosynthetic route, and galvanic replacement reaction [8,13,14]. Driven by the desire to achieve the ultimate goal of clinical translation, the sub-5 nm to 5 nm nano-constructs have been developed to improve the biocompatibility, delivery, and pharmacokinetics of imaging probes and drug delivery systems, as well as in vivo theragnostic and catalysis applications [25,26,27]. The emerging studies have provided new findings that demonstrate the unique size-dependent physical properties, physiological behaviors, and biological functions of the nanomaterials in the range of the sub-5 nm scale, including renal clearance, novel imaging contrast, and tissue distribution. Considering these advances, sub-5 nm Ag–Au nanoalloy synthesis and applications have received significant attention [25,26,27]. However, in sub-5 nm nanoparticles or nanoalloys, synthesis protocols are typically carried out in an organic or biphasic system. Even in an aqueous medium, the methods invariably use thiols, phosphines, or some polymeric capping agents to achieve this size range. Many of these routes have used variety of chemicals for reducing and surface stabilizing that might not be ideal for some biological applications. Additionally, post-synthesis functionalization strategies are essential for the targeted delivery of therapeutic molecules, which could add additional complications, limiting the full potential of mono-/bimetallic NPs [28,29,30,31,32,33,34,35] and making them less feasible for the targeted bio-systems [36]. In these circumstances, the researchers prefer the green chemistry route to overcome undesirable effects and enable opportunities to produce bio-friendly NPs using amino acids, biocompatible polymers, plant extracts, metabolites, monosaccharides, and sucrose [37,38,39,40,41,42]. Previously, various synthesis approaches have been established to fabricate bimetallic Ag–Au alloy NPs, including co-reduction, digestive ripening, laser ablation, and galvanic and anti-galvanic replacement. However, recent challenges involved controlling the size and composition ratio of two or more metals rather than those of their monometallic counterparts. This has led to different mixing patterns, such as core–shell, multi-shell, and intermixing, based on chemical ordering and geometric structure. In all these contexts, there is a need for a green chemistry route to produce biofriendly alloy nanomaterials suitable for biological applications.

The present article reports a single-step and large-scale photochemical synthesis of uniform sub-5 to 5 nm sized AgNPs, AuNPs, and Ag–Au bimetallic alloy NPs in an aqueous L-tyrosine solution. For synthesizing the bimetallic alloy NPs, we selected various percentages (by weight) of Ag–Au compositions (AgAu-25, AgAu-50, AgAu-75), as both the metals are miscible in the bulk phase owing to a practically identical lattice constant. Moreover, following the metal ion photoreduction, the optical, surface charge, hydrodynamic radius, morphology, chemical, and elemental analysis of the synthesized NPs were performed with various characterization techniques. Further, we evaluated the resulting mono-/bimetallic NPs for biocompatibility through cytotoxicity or cell-viability assay and, finally, in vitro peroxidase nanozyme activity. Overall, this strategy provided a facile and environment-friendly route for synthesizing bimetallic alloys, which could have significant potential as a new mono-/bi-metallic nanomaterials class for biomedical and catalysis applications.

## 2. Materials and Methods

### 2.1. Reagents

L-tyrosine, silver sulfate (Ag_2_SO_4_), chloroauric acid (HAuCl_4_), potassium hydroxide (KOH), hydrochloric acid (HCl), and nitric acid (HNO_3_) were purchased from Sigma Aldrich; gold (1000 ppm, Cat. code E3AU4) and silver (1000 ppm, Cat. code E3AGA) element reference solutions for ICP-AES analysis were purchased from Romil Ltd. and used without any purification. Ultrapure deionized water (Millipore water purification system, 18.2 MΩ × cm at 25 °C) was used to prepare all the stock solutions. Entire stock solutions were freshly prepared. The glassware and magnetic stirring bars were washed thoroughly with aqua regia (HCl (37%):HNO_3_ (65%) in a 3:1 volumetric ratio).

### 2.2. Photochemical Synthesis of Monometallic AgNPs and AuNPs

For synthesis, we used a Laboratory Reactor System II for the photochemical synthesis of Ag–Au bimetallic NPs (UV Consulting Peschl). A light source consisting of a medium-pressure Hg vapor lamp (Heraeus TQ 150 W UV lamp) was fitted inside a quartz tube and equipped with a quartz cooling jacket and glass reactor (maximum capacity ~700 mL) [39,40,43]. A 6.5 mL pre-cooled solution of 0.1 M KOH (1.0 × 10^−3^ M) was added to the pre-cooled solution of L-tyrosine (32.5 mL, 2.5 × 10^−3^ M) and Ag_2_SO_4_ (650 µL, 1.0 × 10^−4^ M). The mixture was continuously stirred with a magnetic stirrer. UV light irradiation was performed for 30 min, maintaining the reaction temperature at ~5 °C in the dark (the reaction set-up (glass reactor) was covered with aluminum foil to avoid UV light exposure). The AuNPs were synthesized similarly to the AgNPs except for maintaining the final molar ratios of L-tyrosine:HAuCl_4_:KOH to 30:1:10 and a shorter UV light exposure time (20 min).

### 2.3. Photochemical Synthesis of Bimetallic AgAu NPs

In the alloy synthesis procedure, a pre-cooled solution was prepared consisting of the L-tyrosine (2.5 × 10^−3^ M), metal ions (both Ag^+^ and Au^3+^ ions together having a final conc. of salt 1.0 × 10^−4^ M), and 6.5 mL of freshly prepared 0.1 M KOH (final conc. of 1.0 × 10^−3^ M). The mixture was continuously stirred with a magnetic stirrer under UV light irradiation for half an hour at ~5 °C in the dark. In a total metal ion (Ag^+^ and Au^3+^) concentration of 1.0 × 10^−4^ M, the mole fractions of gold ions (χ_Au_) were 0.25, 0.50, and 0.75, respectively. Due to irradiation with the UV lamp, the Au and Ag ions were simultaneously reduced and formed alloy NPs. Finally, each monometallic or bimetallic NP suspension was concentrated on a rotary evaporator to the desired concentration.

### 2.4. Column Chromatography-Based Purification of the AgNPs, AuNPs, and AgAu NPs

The unbound tyrosine and its photo products and/or any aggregates formed during the photoreduction were separated by column chromatography. Briefly, highly concentrated mono-/bimetallic NP solutions were passed through the column packed with Sephadex G-75 and Sephadex G-25, respectively, and eluted with ultrapure deionized water. All the fractions were collected in separate glass tubes and tested for the presence of NPs. Finally, size measurements of each fraction were performed on a Zetasizer (Nano ZS, Malvern Panalytical Ltd., Malvern, UK). The pre-analyzed fractions of NPs were collected, concentrated, and used for further studies.

### 2.5. Physicochemical Characterization of Monometallic and Bimetallic AgAu NPs

The tyrosine-reduced and column-purified AgNPs, AgAu-25, AgAu-50, AgAu-75, and AuNPs nanoparticles were carefully characterized by analytical, spectroscopic, and microscopic techniques to determine their size, surface charge, morphological and optical features, and chemical/elemental nature. Before performing the UV-visible and DLS measurements, all the purified NP samples were washed with Milli-Q water (5 × 4 mL) and filtered through a 3 kDa Amicon Ultra-4 Centrifugal Filter (Millipore, Burlington, MA, USA) by centrifugation to remove any excess KOH and tyrosine. Specifically, the concentrated pellet or lyophilized powder was used for FTIR, XRD, XRF, and TEM analyses. The UV-Vis spectra and time-dependent absorption changes of the NPs were analyzed by a Cary 300 Bio double-beam spectrophotometer using a 5 mm path length quartz cuvette (1 nm resolution). The hydrodynamic diameter and ζ-potential of the purified aqueous particles were measured by a Nano-ZS DLS device (Malvern Panalytical with 4.0 mV He-Ne 633 nm laser), and nanoparticle tracking analysis (NTA) instruments (Nanosight, Malvern Panalytical), respectively.

#### 2.5.1. X-ray Powder Diffraction

For XRD, samples were prepared by drop-casting the concentrated aqueous NP solution on a cleaned silicon wafer and drying in the air. The XRD measurements were carried out on a Panalytical X’Pert PRO diffractometer with a Cu Kα (1.54 Å) beamline. To determine the size of the Ag, AuNPs, and Ag-Au alloy NPs, the XRD data was analyzed using ‘Origin 8’ software. The (111) Bragg reflection peaks were fitted using the in-built Lorentzian equation to determine the peak width, which was further used in the size calculations according to the Scherrer equation.

#### 2.5.2. Fourier-Transform Infrared Spectroscopy (FTIR)

For capturing the FTIR spectra, the KBr pellets of the pure tyrosine, lyophilized NP samples were prepared using the PIKE CrushIR hydraulic press and dried well under a vacuum desiccator overnight in the dark. The FTIR spectra were recorded with a Bruker Vertex-70 FT-IR spectrometer (scan rate: 60 scans per second).

#### 2.5.3. X-ray Fluorescence Spectroscopy (XRF)

The XRF technique determined the quantitative elemental analysis, purity, and percent stoichiometry of the synthesized mono-/bimetallic NP powder. The NP powder was directly cast on a silicon wafer to form a multilayer film. The XRF measurement was performed using an M4 TORNADO Micro-XRF spectrometer (Bruker Nano, Berlin, Germany) operating at 50 kV/600 µA (30 W), equipped with an X-Flash solid-state silicon drift detector (SSD) having an effective area of 30 mm^2^ with a resolution of 135 eV. Advanced polycapillary X-ray optics were used, which reduces the X-ray spot size to 25 µm, ensuring very high excitation intensity. The data acquisition was performed at different positions on the coated film with an accumulation time of 60 s at each point.

#### 2.5.4. Transmission Electron Microscopy (TEM)

TEM images were acquired using a JEOL Jem 1011 transmission electron microscope (TEM) operated at an accelerating voltage of 100 kV. The solutions containing the NPs were dropped onto a standard carbon-supported mesh copper grid (Formvar/Carbon 300 Mesh Cu) and dried in the dark. The NP size statistical distribution was measured on ~500 NPs, analyzing several TEM pictures and fitted by a normal Gaussian function using ImageJ software Version 1.53t [https://imagej.nih.gov/ij/download.html (accessed on 23 September 2022)].

#### 2.5.5. Inductively Coupled Plasma-Atomic Emission Spectrometry (ICP-AE

The ICP-AES analysis was performed on an ICP-OES Agilent 720/730 spectrometer to determine the Ag and Au ion concentration percentages in the bimetallic alloy NPs. The resulting solution was directly analyzed by an ICP-AES spectrometer. Each Ag–Au alloy NP batch was measured in triplicate (three independent alloy NP dissolutions in aqua regia). The 10–30 µL of alloy NP solution was digested with aqua regia (HCl (37%):HNO_3_ (65%) 3:1) overnight and then diluted to the final desired volume with Milli-Q water. The concentration was determined by comparison with a calibration curve (5–7 concentration levels) obtained from a silver and gold standard solution of 1000 ppm (Romil Pure Chemistry, Cambridge, UK). Measurements were performed following analysis at 3 different wavelengths (for Ag λ = 338.287, 241.310, and 328.068 nm; for Au λ = 208.207, 242.794, and 267.594 nm).

### 2.6. Cell Culture

Breast cancer cells (MCF7) (ICLC number: HTL95021) were maintained in 75 cm^2^ flasks (Sarstedt, Hildesheim, Germany) in high-glucose Dulbecco’s Modified Eagle Medium (DMEM), supplemented with 10% (*v*/*v*) fetal bovine serum, 1% (*v*/*v*) 10,000 U/mL Penicillin and 10,000 U/mL Streptomycin, and 2 mM L-glutamine +1% non-essential amino acids [44]. For all the experiments, the cells were maintained under standard cell culture conditions (5% CO_2_, 95% humidity, and 37 °C temperature) and harvested every 3 days. The cells were passaged at ~80% confluency using a 0.25% (*w*/*v*) trypsin solution containing 0.04% (*w*/*v*) EDTA. Next, the cell lines were incubated with NPs at final concentrations of 0.5 µg/mL, 1 µg/mL, and 5 µg/mL for 24 h at 37 °C in 5% CO_2_. After 24 h of incubation, the samples were washed with PBS pH 7.4, harvested, and then fixed in buffered 3.7% paraformaldehyde for 20 min.

### 2.7. Cell Cytotoxicity Assays

The cell toxicity of the MCF-7 cells was measured after 24, 48, and 96 h of exposure to 0.5, 1.5, and 5 µg/mL of both Ag and AuNPs. We then used colorimetric assays for the detection of highly water-soluble tetrazolium salt WST-8 (2-(2-methoxy-4-nitrophenyl)-3-(4-nitrophenyl)-5-(2,4-disulfophenyl)-2H-tetrazolium, monosodium salt) (Cell Counting Kit-8, Fluka, Sigma-Aldrich, St. Louis, MO, USA). These assays were performed in 96-well plates (Sarstedt) each time. The cells were seeded in microplates at 10,000 cells per well and cultured for 24, 48, and 96 h. Different amounts of Ag, Au, and Ag–AuNPs dispersed in the cell culture medium stock solution were added into different wells, obtaining final NP concentrations of 0.5, 1, and 5 nM. A final concentration of 5% DMSO in the medium was used as the positive control for both cell lines; eight replicates were performed for each investigated point, including the controls and blanks (medium only). An aliquot of WST-8 solution was added to each well and incubated for 3 h in a humidified atmosphere of 5% CO_2_ at 37 °C. Subsequently, the orange WST-8 formazan product was measured at a wavelength of 460 nm in a FLUOstar Optima (BMG LABTECH, Ortenberg, Germany) microplate reader. The data were collected by control software and elaborated with MARS Data Analysis Software Edition8, V5.70 R6 (BMG LABTECH). To express the cytotoxicity, the average absorbance of the wells containing the cell culture medium without cells was subtracted from the average absorbance of the solvent control, 5% DMSO, or nanoparticle-treated cells.

### 2.8. Quantitation of the Intracellular Reactive Oxygen Species (ROS)

The intracellular ROS generation was quantified using an OxiSelect Intracellular ROS Assay Kit. The MCF-5 cells were seeded in 96-well plates at a density of 1 × 10^3^ cells/well. The cells were incubated with 0, 0.1, 0.5, and 5 μg/mL nanoparticles, followed by washing with Hank’s Balanced Salt Solution. Next, the 10 μM dichlorofluorescein diacetate (DCFH-DA) was added to the cells at 37 °C for 1 h in the dark. Nonfluorescent DCFH-DA is converted to fluorescent dichlorofluorescein in proportion to the amount of ROS in the cells. The fluorescence signal was quantified using a spectrofluorometer (DTX800, Beckman Coulter, Inc., Brea, CA, USA) at excitation and emission wavelengths of 485 nm and 530 nm, respectively.

### 2.9. In Vitro Peroxidase Nanozyme Activity of NPs

The in vitro peroxidase nanozyme activity of the synthesized NPs was assayed upon hydrogen peroxide-mediated oxidation of a chromogenic substrate 3,3′,5,5′-Tetramethylbenzidine (TMB). For the testing of in vitro peroxidase nanozyme activity of monometallic Ag and AuNPs, 100 μg/L of metal content concentration was used on the MCF-7 cells. However, in the case of the bimetallic nanoparticles, both the Au and Ag fractions varied successively in AgAu-25, AgAu-50, and AgAu-75 NPs. Typically, a 200 μL aliquot of the above-mentioned NPs was mixed with TMB in the presence of H_2_O_2_ for 400 sec to evaluate the in vitro peroxidase nanozyme activity. As the reaction progressed, the appearance of the blue color was monitored at 650 nm wavelength and room temperature in a real-time kinetic setting using the UV-visible spectrophotometer.

## 3. Results and Discussion

This manuscript reports a facile, single-step, photochemical approach for the green synthesis of ~5 nm monometallic and bimetallic alloy NPs protected with a bio-friendly L-tyrosine (Tyr). As depicted in Figure 1A, we made batches of AuNPs and AgNPs along with three different derivatives of Ag–Au alloy by using different mole fractions (χ_Ag/Au_) of precursor metal ions [Ag^+^]:[AuCl_4_^−^] including 75:25 (Ag-rich, AgAu-25), 50:50 (Equimolar Ag:Au, AgAu-50), and 25:75 (Au-rich, AgAu-75), respectively. The schematic of the photochemical synthesis pathway for the tyrosine-capped NPs is shown in Figure 1B. Mechanistically, under alkaline conditions and UV irradiation, tyrosyl radicals are formed, due to the release of the solvated electron from the phenoxide group of tyrosine [45,46]. These solvated electrons have a high probability of reducing the Ag ions [47]. The tyrosyl radicals undergo crosslinking to yield dityrosine or even higher tyrosine adducts such as trityrosine, tetratyrosine, and pulcherosine [48,49,50]. A pictorial apparatus in Figure 1C represents the assembly of the UV reactor system equipped with the Heraeus TQ 150 W UV lamp used in the NP preparation. Several past studies, including ours, have reported that the nitrogen/sulfur or carboxylic group of the capping (or reducing) agents binds to the surface of the Ag and AuNPs, respectively (Figure 1D). Strong interaction between tyrosine and the surface atoms of NPs leads to the formation of small-sized alloy NPs stabilized by tyrosine, preventing their growth and aggregation [35,40,51,52].

After the NPS and alloy synthesis, UV-visible, ICP-AES, DLS, zeta potential (ζ), XRD, FTIR, XRF, and TEM characterizations were performed to determine the physicochemical characteristics and morphology of the synthesized NPs (Figure 2A). The digital picture of resultant NPs has shown a notable color change in the alloy NPs, with varying Au mole fractions. First, we collected the UV-visible absorption spectra of the resulting monometallic NPs and bimetallic alloy NPs with varying initial Ag or Au molar ratios (Figure 2B). A relatively sharp absorbance peak for the AgNPs (curve a) was observed at 407 nm, whereas the AuNPs (curve e) displayed a broad absorption band around 510 nm. Both bands are attributed to the plasmonic resonance of Ag and Au, respectively [11,53,54]. The absorption feature of the AuNPs compared to the AgNPs is very weak, with a much narrower plasmon line width [55]. The absence of a distinct and intense absorbance band of the AuNPs was observed, as the molar extinction coefficient of the AgNPs is approximately 4 times higher than the AuNPs of similar size [10]. In addition, other absorption bands observed at 430 nm (curve b), 462 nm (curve c), and 488 nm (curve d) correspond to the plasmonic band of the Ag–Au alloy with varying Au mole fractions (χ_Au_) of 0.25, 0.50, and 0.75, respectively. Plotting the maximum wavelength of the absorption band (λ_max_) showed a linear relationship (R^2^ = 0.99) with an increasing percentage of gold mole fractions (χAu) (Figure 2C), accompanied by a redshift in the UV peaks (Figure 2B).

The bimetallic NPs are classified into core–shell, heterostructure, and intermetallic or alloyed nanostructures [1,56]. UV-visible spectra conveniently distinguish two types of bimetallic alloy NPs. In the case of metal–metal core–shell NPs, there are usually two separate peaks referring to the plasmon bands of the separated metal clusters as a core or shell. To further dissect the type of the bimetallic NPs, the UV spectra of the equimolar mixture of Ag and AuNPs were taken, which displayed two distinct SPR bands (Figure 2D). However, the absence of two or more bands in the alloy synthesis also ruled out the possibility of a physical combination of monometallic Ag and Au NPs or bimetallic core–shell NPs. These UV spectral features collectively suggest that the prepared Ag–Au nanoparticles are neither a simple mechanical mixture of individual Ag and Au NPs nor a bimetallic ‘core–shell’ structure. However, they belong to homogeneous alloyed NPs [57,58,59] or a combination of Ag and AuNPs [60].

To understand the formation of alloy NPs, we also performed a time-dependent UV kinetic study on all the mono- and bimetallic NPs (Appendix A). The formation rates of Ag, Au, and Ag–Au bimetallic alloy NPs were measured by observing variations in their plasmon bands at corresponding absorption maxima with time. As shown in Figure 2E, the average time required to complete the formation of AgNPs and the other three Ag–Au alloys was ~10–12 min, while the time for the formation of AuNPs was reduced to 8 min. This indicated that the formation of monometallic AgNPs and bimetallic alloy NPs was slightly slower than AuNPs. Later, we performed the ICP-OES analysis to calculate the percentage of Ag and Au ion concentrations (Figure 2F). The ICP data revealed a close consistency between the atomic percentage (i.e., composition) of the Ag:Au ions that were used in the synthesis (stoichiometrically) and those calculated using ICP.

The DLS measurement showed the hydrodynamic diameter in the range of 7–10 nm with the polydispersity index (PDI) between 0.15–0.2, suggesting the NPs were stable and monodispersed without any visible aggregation (Figure 3A). Next, we acquired the zeta (ζ) potential to confirm the magnitude of the surface charge of all the NP samples (Figure 3B). We observed that the residual surface charge of the NPs becomes negative due to the deprotonation of the surface-bound carboxyl group. From the acquired zeta (ζ) potential data, the magnitude of the residual surface charge on bimetallic metallic NPs becomes higher than the monometallic counterparts (Table 1). This is probably due to the combinatorial surface charges from the precursor Ag and Au nanoalloys.

Furthermore, the XRD and FTIR analyses were executed to verify the successful photoreduction of precursor metal ions to crystalline bimetallic alloy NPs. The XRD pattern of different compositions is shown in Figure 3C. It is observed that AgNPs show four Bragg peaks positioned at 2θ = 38.26°, 44.13°, 64.69°, and 77.70° corresponding to the (111), (200), (220), and (311) lattice planes of standard fcc silver (JCPDS 04-783). The AuNPs also showed similar Bragg peaks at 38.40°, 44.17°, 64.54°, and 77.35°, which corresponds to metallic gold (fcc, #JCPDS 04-784). The XRD results clearly confirmed the successful reduction of the precursor metal ions and the formation of crystalline metallic Ag and AuNPs, respectively. In each case, we observed the broadening of the Bragg diffraction peak in the XRD pattern that confirmed the presence of a very small-sized NP [61]. Additionally, we characterized the Ag–Au alloy samples: the XRD pattern of the alloy samples showed Bragg peaks (Figure 3C, dotted vertical lines) at around 38.50°, 44.18°, 64.76, and 77.80° corresponding to the (111), (200), (220), and (311) lattice planes of standard face-centered cubic (fcc) structures, which indicate that the synthesized NPs were crystalline.

In the XRD results, the broadening of diffraction peaks was observed in the case of the alloy NPs, indicating the formation of smaller-sized bimetallic alloy NPs. It is not possible to determine the Ag–Au alloy formation from the XRD pattern, as silver and gold have the same crystal structure (fcc) and similar lattice constant (0.409 nm vs. 0.408 nm) [62,63]. Therefore, to ensure the Ag–Au alloy NP formation, UV and other characterization techniques were performed. The particle size of the AgNPs, AuNPs, and Ag–Au bimetallic NPs was also calculated from these XRD patterns. The (111) Bragg diffraction peaks were fitted using the in-built Lorentzian equation to determine the peak width and NP size calculated from this value using the Scherrer equation.
$$D_{hkl}=\frac{K\lambda }{\beta_{hkl}\cos\theta }$$
where *D_hkl_* is the apparent crystallite size along the [*hkl*] direction, *K* is the shape factor (*K* = 0.94 in this case), *λ* is the X-ray wavelength, *β_hkl_* is the half-width at half-maximum for the (*hkl*) diffraction peak in radians, and *θ* is the half-scattering angle corresponding to the (*hkl*) diffraction peak. The size of the Ag, AuNPs, and three alloy NPs was calculated using the Scherrer equation in Table 1.

Figure 3D represents the FTIR spectra of tyrosine and tyrosine-capped AgNPs, Ag–Au alloy (varying χ_Au_ 0.25, 0.5, and 0.75 respectively), and AuNPs. In the pure tyrosine FTIR spectra (curve a), the peaks at 1591/1608 cm^−1^ and 1363/1377 cm^−1^ correspond to the asymmetric and symmetric stretching of the carboxylate group [64], respectively. The benzene skeletal CC stretching and the phenol CO stretching are observed at 1514 and 1245 cm^−1^, respectively. In all the NP spectra (curve b–f), a broad tail peak in the range of 1550–1720 cm^−1^ corresponds to the carbonyl stretching region. It appears broader as the percentage of gold mole fraction (χAu) increases compared to that of pure Ag NPs. The peak at 1245 cm^−1^ corresponding to phenolic stretch can be prominently seen without any feature peak in the carbonyl region. In the purified NP samples, similar peaks at 1595, 1514, 1365, and 1245 cm^−1^ (absent in Ag) are observed, confirming dityrosine/polytyrosine molecules as a capping/stabilizing agent. Polymerized forms of tyrosine excel as nanoparticle stabilizers due to their larger size, multivalent binding, adaptable conformation, and improved solvation, which collectively minimize aggregation by providing effective steric hindrance and enhanced surface coverage [48,49,50]. Furthermore, a characteristic peak at 1514 cm^−1^ corresponding to phenolic C-C stretch is almost absent in gold NPs. Similarly, the peak at 1245 cm^−1^ belongs to the phenolic C-O stretch almost absent in the AgNPs case, while in the AuNPs case, it slightly shifted to 1257 cm^−1^. In some cases, a peak at 1455 cm^−1^ is a characteristic of tyrosine and polytyrosine. The distinguishing functional group characteristics peaks (vibrational frequencies), i.e., AgTyr and AuTyr NPs, showed frequencies at ~3200 cm^−1^ and ~1600 cm^−1^ for N-H stretching, and the carbonyl group C=O confirmed the presence of the tyrosine as a capping agent. Thus, it concluded that all these mono- and bimetallic NPs are capped by tyrosine [48,49,50].

Further, the compositions with respect to the atomic ratios of Au:Ag in the bimetallic alloy NPs were verified with XRF spectrometry. Figure 4A–C shows representative XRF spectra of the bimetallic Ag–Au alloy NPs. The spectra A, B, and C correspond to the alloy NPs with gold ion mole fractions values (χ_Au_) of 0.25, 0.50, and 0.75, respectively. After using the Ag and Au signals, the Ag:Au atomic ratios for all the alloy materials were estimated. The compositions estimated from the XRF spectroscopy (Table 1) showed a close match (±5%) with the nominal values and the composition used for the alloy preparation.

To confirm the morphology of all the synthesized NPs, we performed TEM imaging (Figure 5A–E) and analyzed the size distribution of the obtained NPs (Figure 5F–J). In Figure 5A, the AgNPs were mainly less than 6 nm in size. We calculated the size distribution of the AgNPs from the TEM image. From the Gaussian fit of the histogram, the average particle size and its standard deviation were calculated to be around 4.0 ± 2 nm (Figure 5F). On the other hand, the TEM image of the AuNPs (Figure 5E) showed a mean particle size of 3.3 ± 1 nm (Figure 5J). The TEM images in Figure 5B–D present the size and morphology, and the histogram images in Figure 5G–I depict the size distribution of the Ag–Au alloy NPs with varying gold mole fractions of (χ_Au_) 0.25, 0.50, and 0.75, respectively. The TEM acquisitions (Figure 5) show that the particles were well dispersed. By measuring the size of each individual nanoparticle, the mean particle size of the mono- and bimetallic alloy NPs was calculated and is displayed in Table 1. We observed NPs of very small size with narrow-sized distribution. The as-prepared Ag–Au alloy NPs were also small-sized (~5 nm) and had excellent stability, which means they could be stored at room temperature for over a month. The values of the average particle size of the NPs determined by the TEM were in close agreement with the corresponding values determined using the Scherrer equation (Appendix A).

After the detailed physicochemical characterization of the monometallic and bimetallic alloy NPs, we determine the path cellular modifications by assaying various toxicological responsive parameters in the breast cancer cell line (MCF-7) following treatment of the synthesized NPs. Past research has demonstrated the potential of Au–Ag nanoalloys as imaging probes for blood pool imaging and breast cancer screening [65]. Keeping this application in mind, we chose MCF-7, a well-characterized breast cancer cell line, for in vivo studies. We tested tyrosine-capped AgAu NPs to unveil their biocompatibility, in vitro peroxidase nanozyme activity, and controlled cytotoxicity and cell viability (Figure 6A). First, the colorimetric MTT assay was performed upon treatingMCF-7 cells with synthesized NPs. Figure 6B shows that all the treated NPs have minimal toxicity effects at lower NPs concentrations—below ~1 µg/mL. Additionally, the Ag–Au bimetallic alloy NPs show low toxicity, confirming the alloy derivatives’ biocompatible nature. AgNPs and AuNPs appeared to have greater toxicity effects at 5 µg/mL dosages. This in vitro data indicates that any concentration below 1 µg/mL of NPs can be used for biological studies. Further, an MTT assay was performed to evaluate the cytotoxic effect (i.e., the viability of cells) upon introducing both AgNPs and AuNPs and their bimetallic alloy derivatives. Figure 6C clearly showed that exposure to a lower dose of AgAu NPs and Ag–Au alloy NPs of 0.1–1.0 µg/mL showed a good viability percentage. The treatment of MCF-7 cells with the higher concentration of AgNPs and AuNPs (5.0 µg/mL) displayed a distinct decline (<3–4 fold) in the number of viable cells. The dose-dependent activity of the tested NPs suggested that, except for the higher ones, all the low-range doses were less toxic compared to the untreated MCF-7 cells. These results also indicated the dose-dependent biocompatibility of the alloy NPs. Next, after the NP treatment of the MCF-7 cells, the oxidative stress (triggered by an imbalance between the production of free radicals and the activity of the antioxidant defense mechanism) was evaluated [65]. As reactive oxygen species (ROS) mediate oxidative stress, the production of ROS levels in the NP-treated MCF-7 cells was calculated using the fluorescence assay. Compared to the three alloy derivatives, high doses (5 µg/mL) of AgNPs and AuNPs showed significantly increased production of ROS in the cells (Figure 6D).

Furthermore, we theorized that the surface capping of environmentally benign tyrosine molecules might provide biocompatibility and biological identity to the capped NPs. This hypothesis was verified by assessing the in vitro peroxidase nanozyme activity. The metal core-dependent real-time kinetic of in vitro peroxidase nanozyme activity of monometallic AgNPs and AuNPs was determined at comparable metal concentrations. The spectro-colorimetric assay illustrated that the metal AgNPs show significantly increased catalyzing activity of the TMB substrate compared to the Ag NPs, signifying that the Au_100_^Tyr^ NPs have higher in vitro peroxidase-like activity (Figure 6E). Finally, we assessed the equivalent stoichiometric bimetallic alloy nanoparticle derivates (normalized to 100 μg/L concentration for Au with varied Ag fractions) to illustrate the role of Ag–Au elements in the overall in vitro peroxidase activity. Figure 6F reveals that increasing the percentage of Ag in bimetallic alloys can enhance the peroxidase-mimicking behavior incrementally and confirms the role of Ag in TMB catalyzation. Interestingly, the colorimetric assay on both the Ag^+^ and [AuCl_4_]^−^ did not acquire the in vitro peroxidase-mimicking capacity (Appendix A). These outcomes corroborate that only reduced nanoparticulate forms can mimic the in vitro nanozyme-like activity and not the free metal ions [28,35,66,67].

## 4. Conclusions

In summary, we have reported a new and reproducible photoreduction approach for the green synthesis of Ag, AuNPs, and their bimetallic alloy (AgAu NPs) derivatives using tyrosine. This photoreduction strategy was implemented to simultaneously reduce gold and silver salts under alkaline conditions. As a result, this method could produce small-sized alloy NPs with a narrow-sized distribution and excellent stability in an aqueous solution. Following synthesis, the derivatives of the monometallic Ag–Au and bimetallic alloys were characterized by spectroscopic and microscopic techniques (UV-visible, ICP-AES, DLS, zeta potential (ζ), XRD, FTIR, XRF spectroscopy, and TEM) to confirm the physicochemical characteristics and surface morphology. The nano–bio interaction studies performed with the AgAu NPs revealed in vitro peroxidase nanozyme activity and controlled cytotoxicity. The peroxidase nanozyme activity, which is the intrinsic behavior of tyrosine-reduced NPs, depends on the metal core and surface compositions (coatings). Considering these valuable properties, we will test these biocompatible NPs for various biomedical applications, including bioimaging, molecular diagnosis, and drug delivery.

## Figures and Tables

**Figure 1 nanomaterials-13-02471-f001:**
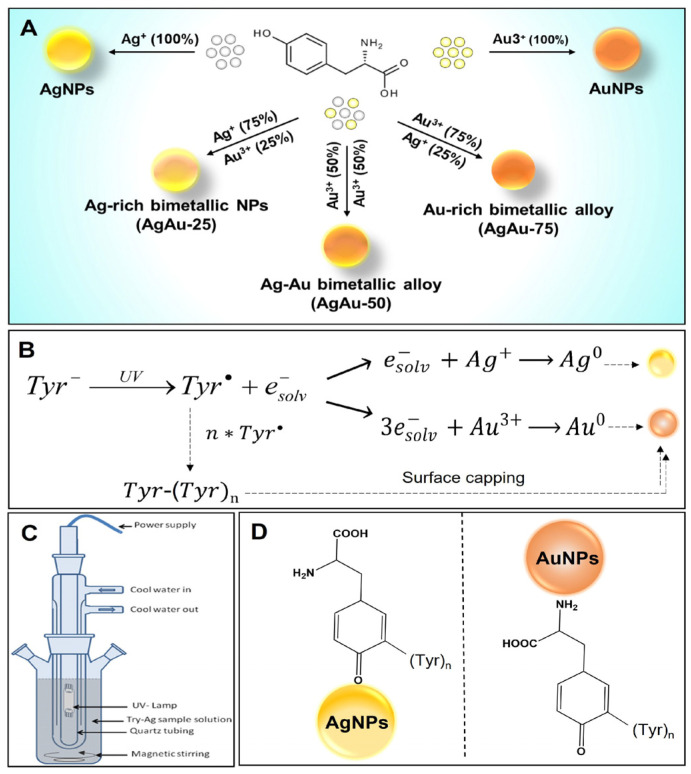
Synthesis scheme and mechanism: (**A**) Schematic presentation of single-step, photochemical synthesis of the mono- and bimetallic NPs of silver (Ag) and gold (Au) using tyrosine (Tyr) as a reducing and stabilizing agent. (**B**) Mechanism of photochemical reduction of metal ions by tyrosine (Tyr)_n_. (**C**) Pictorial diagram of medium-pressure Hg vapor lamp (150 W) set-up. (**D**) The molecular orientation of the tyrosine (Tyr)_n_ on the surface of the AgNPs and AuNPs.

**Figure 2 nanomaterials-13-02471-f002:**
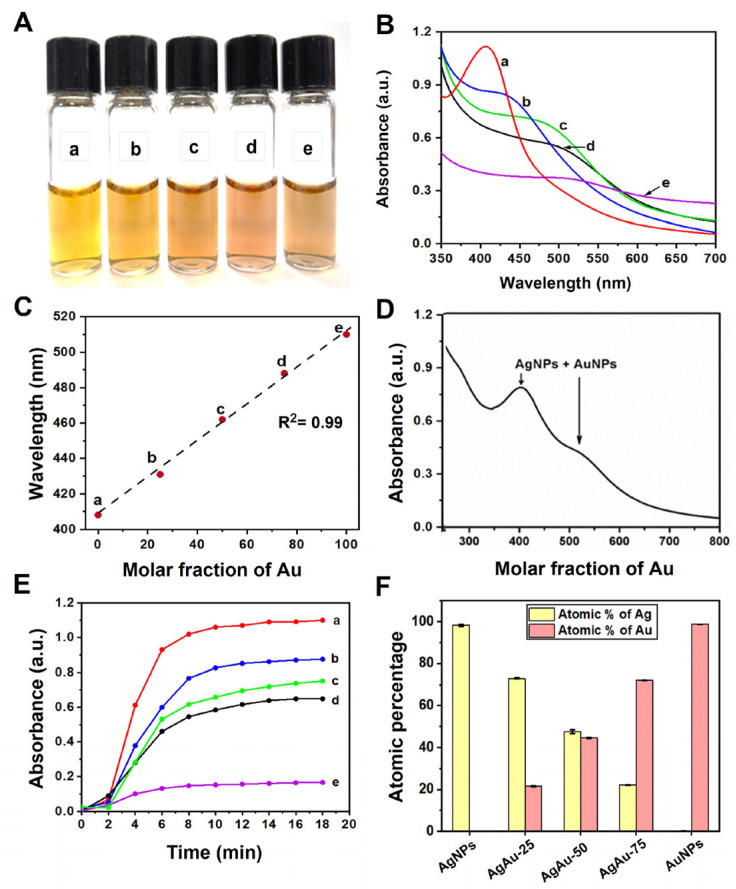
Physicochemical characterization of synthesized NPs. (**A**) Digital photograph of mono- and bimetallic NPs. (a) AgNPs, (b) AgAu-25 NPs, (c) AgAu-50 NPs, (d) AgAu-75 NPs, and (e) AuNPs prepared by photoreduction method. (**B**) UV-visible absorbance spectra of synthesized NPs prepared after irradiation with a medium-pressure Hg vapor lamp. (**C**) The plot of the maximum wavelength corresponding to the SPR band for mono-/bimetallic alloy NPs represents the change in maximum wavelength of the absorption band with the increasing proportion of mole fractions of Au (i.e., χ_Au_). (**D**) UV–vis spectrum of mixture solution of AuNPs and AgNPs in equimolar amount indicating two SPR peaks. (**E**) Absorbance changes for metallic and bimetallic NPs with reaction times at their corresponding absorption maxima. (**F**) The atomic percentage of Ag and Au in monometallic or bimetallic alloy NPs determined by ICP-OES.

**Figure 3 nanomaterials-13-02471-f003:**
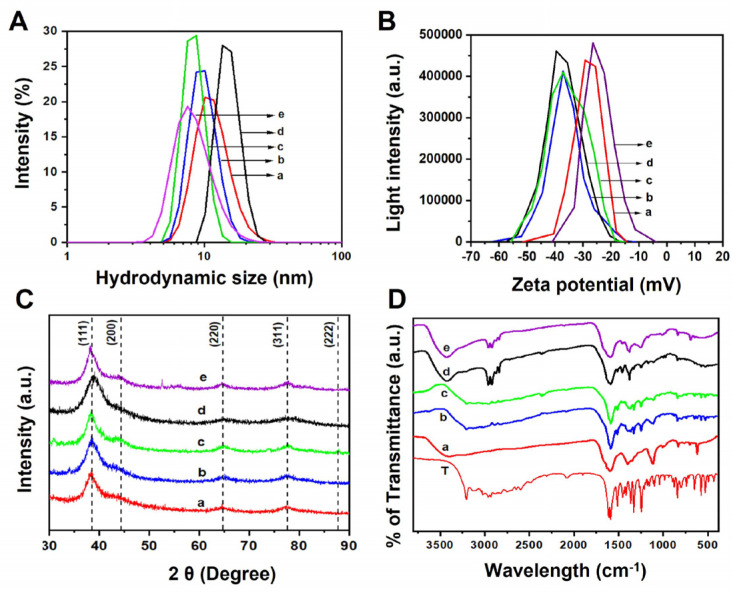
Characterization of the synthesized monometallic and bimetallic AgAu NPs. (**A**) Dynamic light scattering (DLS) spectra measured the hydrodynamic diameter of (a) AgNPs, (b) AuAu-25 NPs, (c) AuAu-50 NPs, (d) AuAu-75 NPs, and (e) AuNPs. (**B**) Zeta (ζ) potential measurement (mV) of synthesized monometallic Ag, Au, and Ag–Au alloy NPs confirmed the magnitude of surface charge after repeated washing and redispersing in distilled water. (**C**) Powder XRD pattern of the Ag, Au, and Ag–Au alloy NPs. (**D**) FTIR spectrum of L-tyrosine (curve T) and synthesized NPs (a–e).

**Figure 4 nanomaterials-13-02471-f004:**
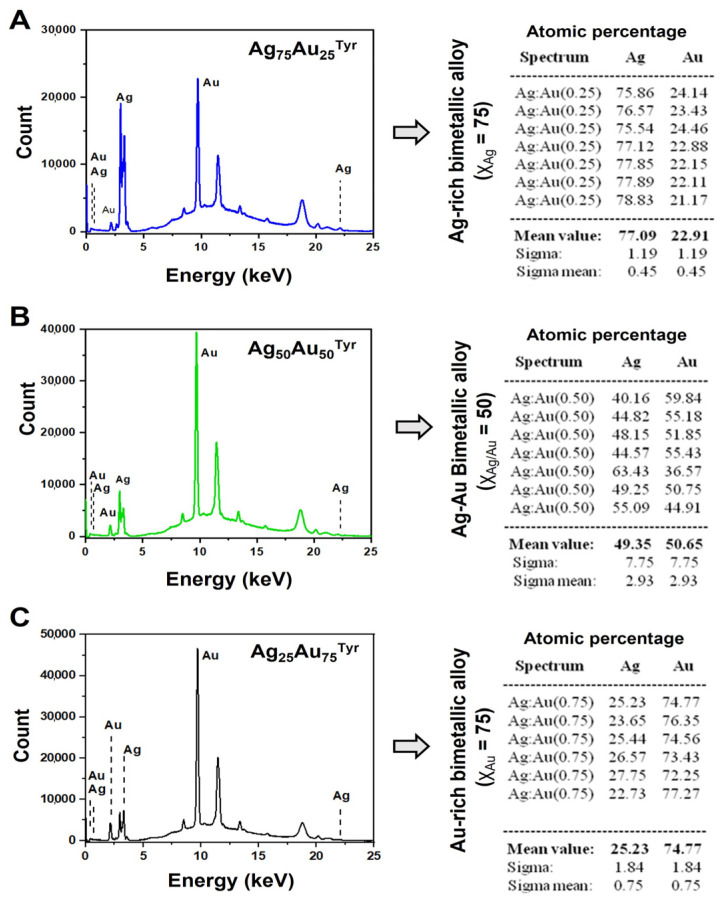
The X-ray fluorescence (XRF) spectra and atomic percentages of Ag and Au metals. The XRF-based quantitative elemental analysis of the Ag–Au alloy NPs: (**A**) AuAu-25 NPs; (**B**) AuAu-50 NPs; (**C**) AuAu-75 NPs. The left column depicts the XRF spectra of bimetallic alloy NPs, and the right column shows the atomic (elemental) percentages of the Ag and Au metals calculated using XRF data. (χ = mole fraction).

**Figure 5 nanomaterials-13-02471-f005:**
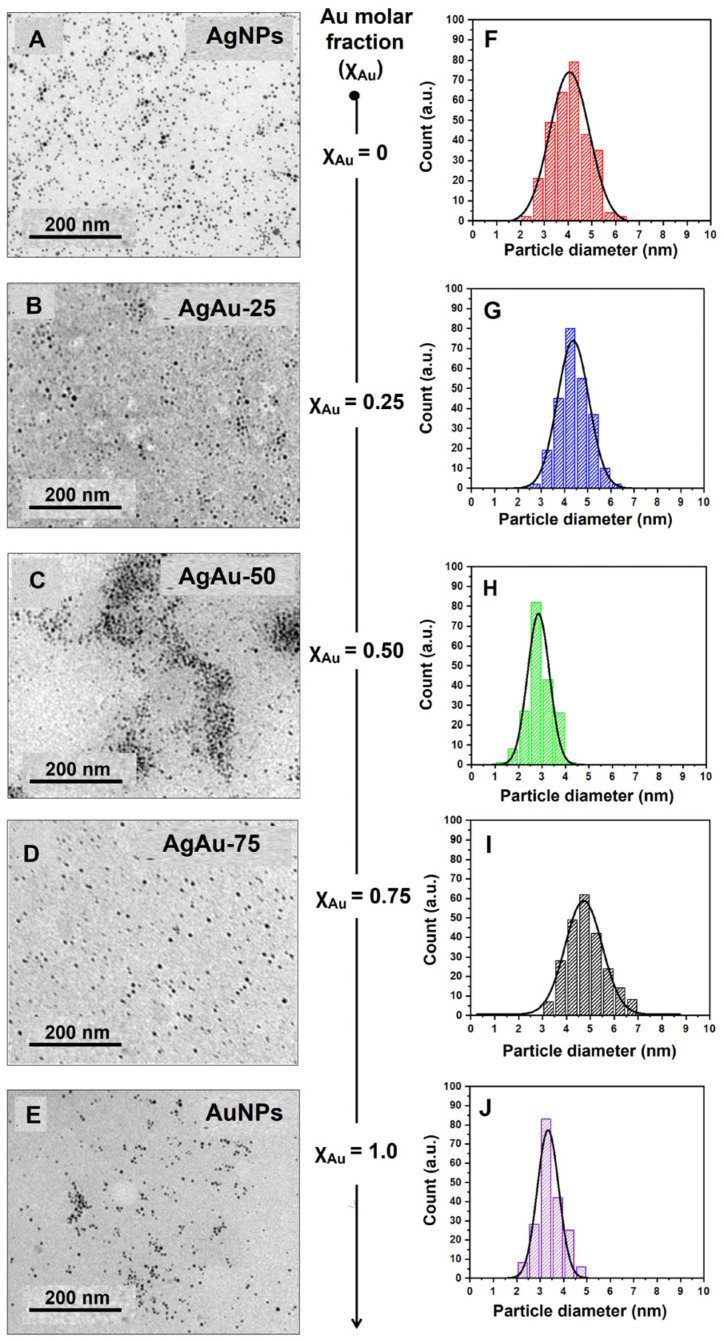
TEM imaging and size distribution analysis. Representative TEM imaging of synthesized NPs (left column) indicating TEM images of (**A**) AgNPs, (**B**) AuAu-25 NPs, (**C**) AuAu-50 NPs, (**D**) AuAu-75 NPs, and (**E**) AuNPs. The right column figures (**F**–**J**) represent the size distribution (SD) histogram of nanoparticle sizes as shown in the images (**A**–**E**), respectively.

**Figure 6 nanomaterials-13-02471-f006:**
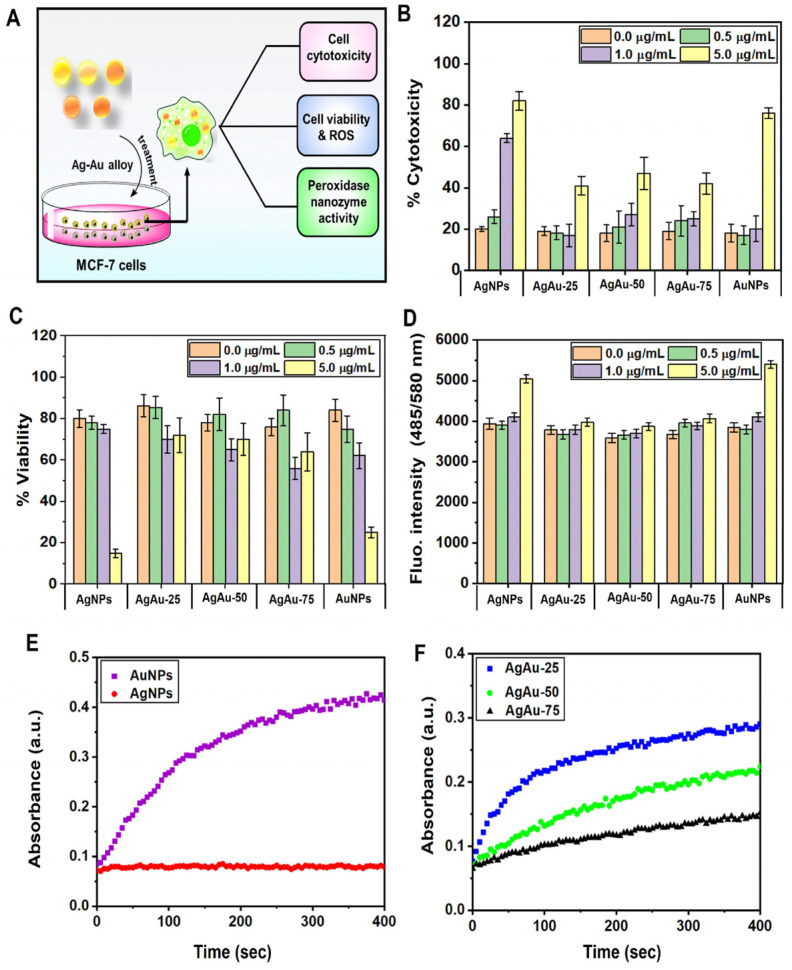
Cell cytotoxicity, viability assay, determination of ROS, and in vitro nanozyme peroxidase activity of tyrosine-reduced mono-/bimetallic NPs. (**A**) Scheme displaying the assessment of the effect of mono-/bimetallic NPs on MCF-7 cells (in vivo). (**B**) LDH assay to determine percentage cytotoxicity in terms of the amount of the released lactate dehydrogenase. (**C**) Percentage viability of the MCF-7 cells calculated by the MTT assay. (**D**) A commercial ROS kit evaluated the intracellular reactive oxygen species (ROS) level in the NP-treated and untreated cells. (**E**) Determination of in vitro nanozyme peroxidase activity of tyrosine-reduced AuNPs and AgNPs, and (**F**) Ag–Au bimetallic NPs with increasing percentages of Au^3+^ ion concentration.

**Table 1 nanomaterials-13-02471-t001:** Summary of spectroscopic characterization of AgNPs, AuNPs, and bimetallic Ag–Au (AgAu-25, AgAu-50, and AgAu-75) alloy NP derivatives.

NPs	χ_Ag_	χ_Au_	UV, SPR ^a^ (λmax, nm)	Zeta [ζ] ^b^ Potential (mV)	Diameter (XRD, nm) ^c^	Atomic % by XRF ^d^	HD (ϕ) ^e^ (DLS, nm)	Diameter ^f^ (TEM, nm)
AgNPs	1.00	0.00	408	−26.0	3.5	-	10 ± 2	4.0 ± 2.2
AgAu-25	0.75	0.25	430	−37.0	3.1	77:23	8 ± 1	4.7 ± 1.8
AgAu-50	0.50	0.50	462	−37.2	3.8	49:51	9 ± 1	3.0 ± 1.5
AgAu-75	0.25	0.75	488	−39.3	3.0	25:75	8 ± 1	4.7 ± 2.1
AuNPs	0.00	1.00	510	−29.2	3.8	-	7 ± 1	3.3 ± 1.9

^a^ Surface plasmon resonance (SPR) measured with UV-Vis spectrophotometer, ^b^ Zeta [ζ] potential value (mV) observed at the highest intensity. ^c^ Average particle diameter determined by X-ray diffraction (using Debye–Scherrer equation), ^d^ Atomic percentage (quantitative elemental analysis) of Ag and Au calculated using XRF spectroscopy. ^e^ Particle size/hydrodynamic diameter (HD) of synthesized NPs calculated with dynamic light scattering (DLS) instrument. ^f^ Particle **^particle^** size (average diameter) measured by counting more than 200 NPs (TEM images) and using Gaussian curve fitting.

## Data Availability

Not applicable.

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
