# Peer review of "Silver–Gold Alloy Nanoparticles (AgAu NPs): Photochemical Synthesis of Novel Biocompatible, Bimetallic Alloy Nanoparticles and Study of Their In Vitro Peroxidase Nanozyme Activity"

_nanomaterials, 2023, doi:10.3390/nano13172471_

Round 1

Reviewer 1 Report

In this manuscript, authors synthesized bimetallic alloy nanoparticles (AgAu NPs) with different ratios of Ag/Au to study their in vitro peroxidase nanozyme activity. All experimental results had no problems. But some issues need to be addressed. Questions and comments are as below.

1. Authors should confirm that the structure of as-synthesized AgAu NPs is alloy or bimetal. Alloy and bimetal are different and their properties also are different. Using elemental mapping analysis could confirm the structure of AgAu NPs.

2. Authors should give evidence that molecular orientation of the tyrosine on the surface of AgAu NPs. Did the Ag/Au ratio of AgAu NPs affect all experimental results?

3. Authors should detail discuss why Ag NPs and Au NPs had the highest cytotoxicity (Figure 6B) and the lowest viability (Figure 6C) at 5.0 ug/mL as compared to AuAg-25, AgAu-50, and AgAu-75. Does these results mean that the cytotoxicity and viability are not related to Ag? Because the cytotoxicity and viability of AuAg-25, AgAu-50, and AgAu-75 have no obvious difference. Same question is for ROS (Figure 6D).

4. Please correct the figure caption of Figure 6. There are two "(B)".  

5. Figure 6E and 6F could be combined together.

 Minor editing of English language required.

Author Response

Assessment: In this manuscript, authors synthesized bimetallic alloy nanoparticles (AgAu NPs) with different ratios of Ag/Au to study their in vitro peroxidase nanozyme activity. All experimental results had no problems. But some issues need to be addressed. Questions and comments are as below.

Response: We thank the reviewer for their thoughtful comment. We have addressed all the suggested comments. 

Concern 1: Authors should confirm that the structure of as-synthesized AgAu NPs is alloy or bimetal. Alloy and bimetal are different, and their properties also are different. Using elemental mapping analysis could confirm the structure of AgAu NPs.

Response: We thank and appreciate the reviewer’s comment. We agree that the structure of the AgAu can be alloy or bimetal. As indicated in table 1, we perform elemental analysis, which revealed that (111), (200), (220), and (311) lattice planes of standard face-centered cubic (fcc) structures, which indicate that synthesized NPs were crystalline. However, due to unavailability of the elemental mapping analysis in present lab, we were unable to perform the suggested analysis. 

Concern 2: Authors should give evidence that molecular orientation of the tyrosine on the surface of AgAu NPs. Did the Ag/Au ratio of AgAu NPs affect all experimental results?

Response: Thank you for the reviewer's concern. Tyrosine orientation was evidently said by  references we cited, and inference form the IR data (Fig 3D) description on page No. 10. The variable stretch 

Concern 3: Authors should detail discuss why Ag NPs and Au NPs had the highest cytotoxicity (Figure 6B) and the lowest viability (Figure 6C) at 5.0 ug/mL as compared to AuAg-25, AgAu-50, and AgAu-75. Do these results mean that the cytotoxicity and viability are not related to Ag? Because the cytotoxicity and viability of AuAg-25, AgAu-50, and AgAu-75 have no obvious difference. The same question is for ROS (Figure 6D).

Response: We thank the reviewer for the constructive and critical comments and suggestions to improve the quality of the manuscript. We agree with the reviewer's concern and also believe that the cytotoxicity is related to the silver as well. However, the effect of a green capping agent (tyrosine) might have shown comparable toxicity. The formation of gold-silver alloy nanoparticles reduced the leaching of silver ions from the AgAuNPs compared to silver nanoparticles, thereby improving biocompatibility. These (toxicity and ROS) observations were also supported by the article by Nanoscale. 2016 Jul 14; 8(28): 13740–13754 and Nanoscale, 2017, DOI: 10.1039/C7NR04979J and Materials Science & Engineering C 96 (2019) 286–294 

Concern 4: Please correct the figure caption of Figure 6. There are two "(B)". 

 Response: We thank the reviewers for their corrections. We have made appropriate changes in the figure captions of the manuscript file. 

Concern 5:. Figure 6E and 6F could be combined together.

 Response:  We thank the reviewers for their suggestions. We agree with the reviewer that we can combine them, however, we have not clubbed them just to maintain the figure symmetry and also to visualize the effect of a bimetallic and monometallic better way for a head-to-head comparison.    

Reviewer 2 Report

When I started looking at the manuscript, one thing struck me particularly: in most of the Figures, the labels of the axes are badly wrong. It appears the authors haven’t the faintest idea of what those graph report and arbitrarily decided what units to put on the axes. This is totally unacceptable, and I refuse to review a manuscript in which the authors were so careless of not checking what was reported in their graphs. Furthermore, many captions contain errors in the description of the figures and or do not allow the reader to understand correctly what is reported in the graphs. Because of this the manuscript cannot be considered and should be sent back to the authors for proof checking. I will properly review a resubmitted version. The time of the reviewers is precious and cannot be abused. 

English is acceptable. Minore cecking

Author Response

We appreciate the reviewer for their suggestion and apologize for any incorrect framing of the sentences in the figure captions. We have revised the suggested changes in the manuscript accordingly. 

Reviewer 3 Report

Well designed manuscript. The idea is clear.

Conclusions are supported by experimental data

Author Response

We are grateful for the reviewer's appreciation about the design of the manuscript and research idea. 

Round 2

Reviewer 2 Report

Some changes have been introduced in the Figures. However, the following figures (see below) still present problems. I maintain the authors redacted the manuscript in a sloppy way. They were given a chance to correct their mistakes. They did a poor job.

a) List of Figures with major mistakes:

Fig. 2 C: the Y axis is not correct; D: the numbers are missing in the Y axis; E: the label of Y axis is not correct.

The caption of Fig. 2B reads AuAu for all compositions.

Fig. 3. A: caution, dynamic light scattering reported as intensity is not indicative of distribution as intensity depends on size!; B: both labels of the X and Y axes are not correct; no numbers are present in Y axis; D: Transmittance is typically reported as %.

Figure 5 F-J: all Y axes labels are wrong.

Other mistakes: after line 409 the sentence is not completed, and the following text is missing

b) Considerations on the manuscript.

1) The authors do not explain what the source of radicals (ROS) causing the cell damage. They state that the oxidative stress is “triggered by an imbalance between the production of free radicals and the activity of the antioxidant defense mechanism” It is not clear how the nanoparticles operate to trigger such an imbalance. An explanation is required.

2) The mechanism at the basis of the claimed peroxidase-like mechanism is not discussed. In the reference provided, the passivating agents appear to be responsible for such a mechanism. Is tyrosine involved here, too? What is its role (if any)? Incidentally, I could not find a proper reference to nanozymes.

3) On lines 492-493 it is claimed that AuNP are better than AgNP. However, on line 499 it is stated that in the bimetallic alloy, the higher content in Ag provides a higher activity. The role of Ag is not clearly explained and this statement might cause confusion.

4) It is stated (line 403) that polymerized species are better stabilizing agents. This statement needs to be supported in view of the large excess of tyrosine present (30:1 if I understood correctly).

Finally, the English is poor and needs significant improvement.

My conclusion is that this manuscript requires substantial revision both on several basic key points and the formal representation (as highlighted in my original examination of it). I recommend a new version is resubmitted seriously considering the above comments. A detailed (point-by-point) illustration of the changes introduced must accompany this resubmission. Acceptance of the resubmitted manuscript will depend on the result of its accurate review by the referees.

See above

Author Response

Comment 1: Some changes have been introduced in the Figures. However, the following figures (see below) still present problems. I maintain the authors redacted the manuscript in a sloppy way. They were given a chance to correct their mistakes. They did a poor job. a) List of Figures with major mistakes: Fig. 2 C: the Y axis is not correct; D: the numbers are missing in the Y axis; E: the label of the Y axis is not correct. The caption of Fig. 2B reads AuAu for all compositions. Fig. 3. A: caution, dynamic light scattering reported as intensity is not indicative of distribution as intensity depends on size; B: both labels of the X and Y axes are not correct; no numbers are present in the Y axis; D: Transmittance is typically reported as %. Figure 5 F-J: all Y axes labels are wrong. Other mistakes: after line 409 the sentence is not completed, and the following text is missing.

Response: We thank you, reviewer, for pointing out the errors and addressing necessary changes. We apologize for that. We have fixed them and added the revised figure on the tracked changed paper version. In addition, I have also included the missing part after line 409.  

Comment 2: The authors do not explain the source of radicals (ROS) causing cell damage. They state that oxidative stress is "triggered by an imbalance between the production of free radicals and the activity of the antioxidant defense mechanism" It is not clear how the nanoparticles operate to trigger such an imbalance. An explanation is required.

Response: Thank you for your concern. Please find the following explanation for your concern. To the best of our knowledge, there is enough research specifically focused on gold-silver alloy nanoparticles and their role in triggering an imbalance in oxidative stress. However, I can provide a general explanation of how silver and gold nanoparticles may induce oxidative stress.

Silver nanoparticles (AgNPs) and gold nanoparticles (AuNPs) are well-known for their unique physicochemical properties and potential applications in various fields, including medicine and industry. When these nanoparticles interact with biological systems, they can generate reactive oxygen species (ROS) as a byproduct of their surface reactions and electron transfer processes. ROS are highly reactive molecules that include superoxide anion (O2·-), hydrogen peroxide (H2O2), and hydroxyl radicals (·OH), among others. The generation of ROS by nanoparticles can lead to oxidative stress, which is characterized by an imbalance between the production of ROS and the body's antioxidant defense mechanisms. The excessive production of ROS overwhelms the cellular antioxidant capacity, leading to oxidative damage to various cellular components, including lipids, proteins, and DNA. The specific mechanisms through which silver and gold nanoparticles trigger oxidative stress may vary and are still an area of ongoing research. Some proposed mechanisms include:

Direct interaction with cellular components: silver and gold nanoparticles can directly interact with cellular membranes, proteins, and organelles, leading to the generation of ROS and subsequent oxidative damage. Electron transfer and catalytic reactions: These nanoparticles possess a large surface area and unique electronic properties, making them efficient catalysts for redox reactions. In biological environments, they can catalyze the production of ROS from molecular oxygen. (Ref. Int J Mol Sci. 2018 May; 19(5): 1305 Int J Mol Sci. 2018 May; 19(5): 1305, Nanotoxicology.2011 Mar;5(1):43-54).

Induction of inflammatory responses: Nanoparticles can activate immune cells and initiate inflammatory responses. Inflammation can, in turn, lead to the production of ROS by activated immune cells (Ref. Nanoscale Adv. 2022 Aug 11; 4(16): 3300–3308).

Release of metal ions: In the case of silver nanoparticles, the release of silver ions in cellular environments can contribute to ROS generation through Fenton-like reactions. It is essential to note that the biological effects of nanoparticles, including their potential to induce oxidative stress, can be influenced by various factors, such as nanoparticle size, shape, surface coating, concentration, and exposure duration. Moreover, the understanding of gold-silver alloy nanoparticles' specific effects is still an emerging field, and more research is needed to elucidate their precise mechanisms of action in inducing oxidative stress (Ref. Materials (Basel). 2021 Jan; 14(1): 53, Yonsei Med J. 2014 Mar 1; 55(2): 283–291).

Comment 3: The mechanism at the basis of the claimed peroxidase-like mechanism is not discussed. In the reference provided, the passivating agents appear to be responsible for such a mechanism. Is tyrosine involved here, too? What is its role (if any)? Incidentally, I could not find a proper reference for enzymes.

Response: We thank the reviewers for their comments and suggestions. Yes, tyrosine is involved in the peroxidase-like mechanism described in the study. Under alkaline conditions and UV irradiation, tyrosyl radicals are formed due to the release of the solvated electron from the phenoxide group of tyrosine. These solvated electrons have a high probability of reducing the Ag ions. The tyrosyl radicals then undergo crosslinking to yield mono or di-tyrosine or higher tyrosine adducts. This process allows tyrosine-capped nanoparticles to exhibit peroxidase-like activity, mimicking the behavior of natural peroxidase enzymes. The basis of the claimed peroxidase-like mechanism involves the generation of tyrosyl radicals from tyrosine-capped nanoparticles under alkaline conditions and UV irradiation. These radicals can participate in oxidation-reduction reactions, like natural peroxidase enzymes. The tyrosyl radicals can catalyze reactions involving substrates like TMB (3,3',5,5'-tetramethylbenzidine) to produce a colorimetric response, allowing the nanoparticles to act as peroxidase mimics.

Comment 4: On lines 492-493 it is claimed that AuNP is better than AgNP However, on line 499 it is stated that in the bimetallic alloy, the higher content in Ag provides a higher activity. The role of Ag is not clearly explained, and this statement might cause confusion.

Response: Thank you for your concern. We corrected the mentioned lines. Here is a short description of the role of Ag ions in the in vivo peroxidase nanozyme activity. In the context of a gold-silver alloy, the role of the silver ion (Ag+) is to contribute to the enhancement of in vitro peroxidase activity when interacting with the 3,3′,5,5′‑Tetramethylbenzidine (TMB) substrate. The alloy's composition, particularly the presence of silver, influences the catalytic behavior of the alloy nanoparticles. The silver ions in the alloy play a crucial role in promoting the catalytic activity of the alloy, resulting in increased peroxidase-like behavior. This interaction between the silver ions and the TMB substrate is essential for the observed enhancement of the catalytic activity, as evidenced by the significant increase in catalyzing activity with TMB seen in the presence of the gold-silver alloy.

Comment 5: It is stated (line 403) that polymerized species are better stabilizing agents. This statement needs to be supported in view of the large excess of tyrosine present (30:1 if I understood correctly).

Response: Thank you for your concern. A large excess of tyrosine has been taken for the reaction as tyrosine acts as a reducing agent and stabilizing agent for the NPS or alloy NPS. However, related to its polymerized form as a great stabilizing capability, I have the following points for discussion. I have added the description to the original manuscript.

Polymerized species are often considered superior stabilizing agents for nanoparticles due to their unique molecular characteristics and enhanced interaction capabilities. This superiority is especially pronounced when compared to individual monomeric molecules. Here's why polymerized species are better stabilizing agents for nanoparticles: Increased Molecular Size: Polymerized species, being larger molecules formed by the repetition of monomer units, possess a greater molecular size compared to individual monomers. This increased size allows polymerized species to provide a more effective steric hindrance, preventing nanoparticles from coming into proximity and thus reducing the chances of aggregation. Enhanced Surface Coverage: The larger surface area of polymerized species allows them to cover a more significant portion of the nanoparticle surface, forming a denser and more comprehensive protective layer. This layer acts as a physical barrier that shields nanoparticles from interactions that could lead to aggregation, such as van der Waals forces. Multivalent Binding: Polymerized species often have multiple functional groups along their chains. These groups can establish multiple binding interactions with the nanoparticle surface, increasing the strength and stability of the interaction. This multivalent binding provides a more secure attachment of the stabilizing agent to the nanoparticle, reducing the likelihood of detachment and aggregation. Conformational Flexibility: The flexibility inherent in polymerized chains enables them to adapt to various nanoparticle shapes and surface features. This conformational adaptability ensures that the stabilizing agent can effectively cover irregularities on the nanoparticle surface, leading to a more uniform and stable dispersion. Charge Distribution: Polymerized species can have a more evenly distributed charge due to the presence of multiple functional groups. This balanced charge distribution enhances electrostatic repulsion between nanoparticles, preventing them from close contact and subsequent aggregation. Improved Solvation: Multiple functional groups in polymerized species allow for stronger interactions with solvent molecules, resulting in a more robust solvation layer around the nanoparticles. This solvation layer further inhibits aggregation by maintaining a stable dispersion in the surrounding medium.

Comment 6: My conclusion is that this manuscript requires substantial revision both on several basic key points and the formal representation (as highlighted in my original examination of it). I recommend a new version is resubmitted seriously considering the above comments. A detailed (point-by-point) illustration of the changes introduced must accompany this resubmission. Acceptance of the resubmitted manuscript will depend on the result of its accurate review by the referees.

Response: Thank you for your conclusive response. We hope that the revised manuscript checked with Grammarly will meet the expectations of the reviewers and editors and address the concerns raised by the reviewers. We thank the review team for the meticulous review and suggestions, which have greatly improved the manuscript.
